# Format chain exchange (FORCE) for high-throughput generation of bispecific antibodies in combinatorial binder-format matrices

Stefan Dengl[1], Klaus Mayer[1], Felix Bormann[1], Harald Duerr[1], Eike Hoffmann[1], Bianca Nussbaum[1], Michael Tischler[1], Martina Wagner[1], Andreas Kuglstatter[2], Lea Leibrock[2], Can Buldun[1], Guy Georges[1] & Ulrich Brinkmann [1][✉]

Generation of bispecific antibodies (bsAbs) requires a combination of compatible binders in formats that support desired functionalities. Here, we report that bsAb-matrices can be generated by Format Chain Exchange (FORCE), enabling screening of combinatorial binder/format spaces. Input molecules for generation of bi/multi-valent bsAbs are monospecific entities similar to knob-into-hole half-antibodies, yet with complementary CH3-interface-modulated and affinity-tagged dummy-chains. These contain mutations that lead to limited interface repulsions without compromising expression or biophysical properties of educts. Mild reduction of combinations of educts triggers spontaneous chain-exchange reactions driven by partially flawed CH3-educt interfaces resolving to perfect complementarity. This generates large bsAb matrices harboring different binders in multiple formats. Benign bio-physical properties and good expression yields of educts, combined with simplicity of pur-ification enables process automation. Examples that demonstrate the relevance of screening binder/format combinations are provided as a matrix of bsAbs that simultaneously bind Her1/Her2 and DR5 without encountering binder or format-inflicted interferences.

[1] Roche Pharma Research and Early Development (pRED), Large Molecule Research, Roche Innovation Center Munich, Penzberg, Germany. [2] Roche Pharma Research and Early Development (pRED), Structural Biology, Roche Innovation Center Basel, Basel, Switzerland. [✉]email: ulrich.brinkmann@roche.com

Simultaneous targeting of different antigens by bispecific antibodies (bsAbs) can elicit novel functionalities and may enable innovative therapeutic applications. Desired functionalities for bispecific or multispecific antibodies include targeting of two antigens on the same cell surface (for agonistic functionalities, blocking of multiple receptors, and avidity-mediated selectivity effects), two antigens on different cell surfaces (linking of effector and target cells), or two soluble effectors[1–5].

Discovery of bsAbs with such functionalities is a complex task because multiple parameters need to be set correctly to achieve a desired functional outcome. These parameters include (i) properties of the individual binders (on-/off-rates, epitopes), (ii) arrangement of binders in relation to each other and the Fc fragment (format geometry), (iii) distances of paratopes, and (iv) number of valencies of each binder. In addition, overall size (hydrodynamic radius) of the molecule may be relevant[6–10]. The format defines function principle has proven to be relevant in numerous examples. This includes early attempts to inactivate cMET signaling on tumor cells via cMet specific antibodies, which were successful only with monovalent binders and failed when applying bivalent IgG's (bivalency leads to receptor activation rather than inactivation[11,12]). The observation that bivalent formats enhance activation of receptors eliminated such antibody-format combinations for cMET but activation via bivalent binding can be of advantage for other applications. Bi- or multivalency of targeting arms can for example induce intracellular pathways, including tumor-targeted activation of death receptor mediated apoptosis[13]. Vice versa, modification or multiplication of constant regions can modulate or enhance Fc-mediated functionalities[14]. Other examples that highlight the functional importance of the geometry and arrangement of binders include the necessity to combine compatible epitopes and geometries to achieve functional linking of clotting factors[15,16], and observations that blood–brain–barrier-penetration functionalities of bispecific antibodies depend on affinity and valency of their shuttle modules[17–20]. Inhibition of Her2-mediated signaling as a feature of Trastuzumab can be altered to activation of signaling by rearranging its Fab fragments into different configurations. Cross-linking unpaired cysteines on the surface of the heavy and light chains of these Fabs resulted in bivalent Her2-binders with agonistic activity, instead of antagonistic activity of Trastuzumab IgG in regular configuration[21]. Single-domain antibodies against NGF and VEGF show increased functional potency when brought into a specific format configuration, fused to the C-terminus of an IgG[22]. Appropriate binder–format combinations are also important for the generation of tumor-targeted T-cell recruiting bi-specific antibody derivatives (1–4, 9, 23, 24). For example, carcinoembryonic antigen (CEA)-targeting entities have been generated that show improved tumor-targeting in formats bivalent for CEA-binding, and improved efficacy with one of the CEA-binders arranged in a head-to-tail configuration with the CD3-binder[23]. The application of monovalent, low-affinity binding to CD3 leads additionally to prevention of (non-targeted) T-cell activation in the absence of tumor cells. Further examples for format dependent activity modulation/enhancement of T-cell recruiting antibody derivatives are TriFabs or Contorsbody-TriFabs in which the format places a monovalent CD3 binders into close proximity to cell surface binders[24].

Over the past years, bsAbs have been generated and characterized in different formats. This enabled the deduction of some potential rules regarding binder and format requirements. However, it has become clear that bsAbs with optimal functionalities cannot be crafted in general via rational design just by applying those rules. Analyses of different binder combinations in different formats are required for comprehensive coverage of the bsAb

design space. Addressing such a large design space to identify bsAb lead molecules has until now been hampered by limitations in producing large numbers of diverse binder–format combinations via conventional antibody production processes. The development of such techniques is therefore important to increase efficiency and reduce cycle times of bispecific therapeutic antibody discovery. Some technologies already exist for the generation and assessment of binder combinations in standard IgG-formats, including Genentechs knob-into-hole based half-antibodies[25–27] and Genmabs IgG4 based Fab Arm exchange technologies[28–30] and variations thereof[31,32]. These approaches may still pose in some aspects technical challenges (dimers or aggregates in half antibody production; high-throughput monitoring of IgG4 exchange efficacies), but overall can be considered as valuable, sufficiently robust, and state-of-the-art technologies for generating binder-binder matrices in a bivalent IgG format. Considering that format can define or at least heavily impact function, identification of an optimal (preferentially best) bsAb requires not only to assess combinations of different binders. It additionally requires the generation and assessment of binder combinations in different formats because binder pairs that work with one format may not work with another. The comprehensive coverage of such design spaces requires the generation of many entities, even with rather limited numbers of initial binding entities and format choices as input. For example, a combinatorial matrix of just 32 predefined binders against target A and 32 against target B (realistic numbers considering diversity of binding kinetics and epitope coverage of antibodies selected in antibody lead identification campaigns) in just 3 input format combinations requires the production of >9000 molecules to completely cover the design space. Combining the same binders in 5 format combinations would cover >25,000 molecules. It is obvious that covering such large matrices requires robust processes with sufficient simplicity to enable automation.

In this work, we describe Format Chain Exchange (FORCE) as a high-throughput technology that enables efficient combinatorial generation of bispecific antibodies in different configurations for "screening in final format". The method bases on bsAb assembly from monospecific educt molecules harboring different binders in different formats. Assembly occurs via a heavy-chain exchange reactions driven by engineered Fc-dummy chains contained in the monospecific input molecules. Efficacy, robustness and simplicity (incl. production and one-step product purification) enables process automation to enable comprehensive screens of bsAb binder–format design spaces.

## Results

**Dummy–CH3 interface driven format chain exchange**. The overall principle of the FORCE process is shown in Fig. 1. It is based on CH3-interface driven exchange reactions of input molecules representing different binder/format combinations. Educt molecules are knob-into-hole heterodimers of one half-antibody-like productive side and one knob or hole dummy Fc (Fig. 1a). The dummy associates with the knob or hole Fc of the productive side (covalently linked by hinge interchain disulfides) and thereby covers the CH3 interface. The presence of the dummy chain prevents aggregation and/or dimerization frequently observed with half-antibody like molecules that possess free CH3 interfaces. The productive sides have antigen-binding Fab arms attached to KiH-Fc-dummy heterodimers either N-terminal via a regular IgG1-hinge, or C-terminal via flexible (Gly4Ser)n linkers, or N-as well as C-terminal. Different mutations built into the knob or hole dummy chains lead to limited repulsions within CH3-interfaces, yet without affecting expression, purification or biophysical properties of educt molecules

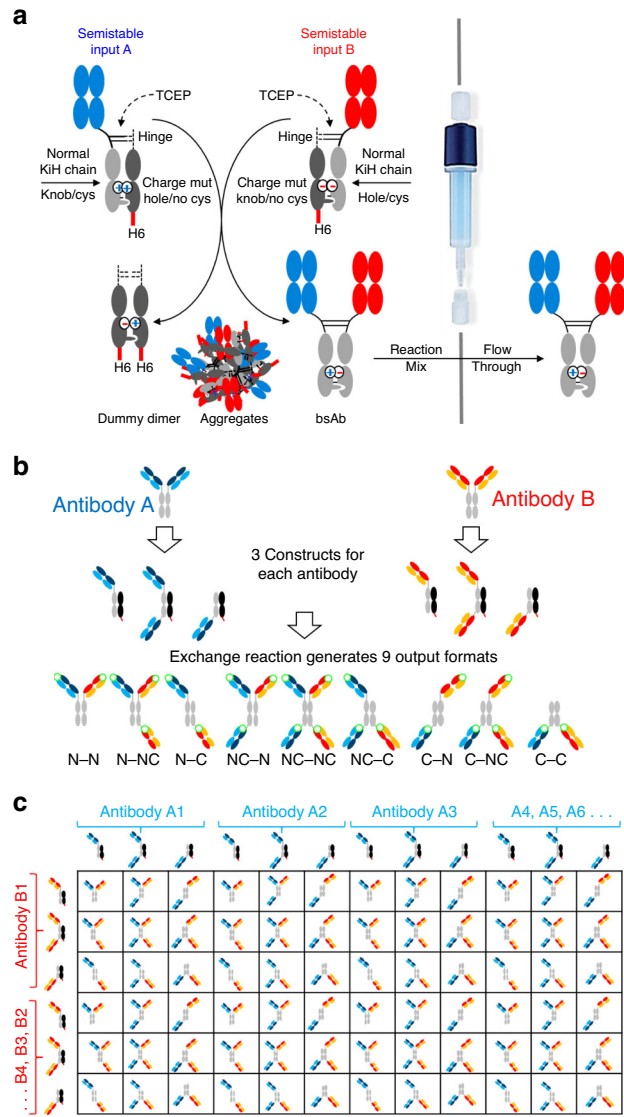

**Fig. 1 Generation of combinatorial binder/format bsAb matrices via dummy-Fc driven assembly. a** Affinity tag-containing dummy chains drive bsAb assembly and enable one-step purification. **b** Combination of educts of different formats generates bsAb format diversity. Green circles indicate the location of paratopes in output formats. **c** Combining binders in different input formats multiplies the binder–format design space. Each product within the matrix is a unique bsAb with a specific binder combination in a specific format.

(described in detail in subsequent sections). Limited reduction of hinge disulfides initiates an exchange reaction between complementary educts driven by their partially destabilized interfaces. The reaction resolves the flawed CH3-educt interfaces to perfect complementarity in the products of the exchange reaction. Those products are bsAbs with regular CH3–CH3 interface-disulfide stabilized knob-into-hole Fc's and dummy-dimers, each with non-repulsing interfaces. Affinity tags attached only to the dummy-chains enables absorbing any remaining educts, dummy dimers and potential aggregates from the exchange reaction. This provides a single-step robust purification step to obtain pure bsAbs.

Knob or hole dummy-chains with exchange-driving mutations can be combined with half-antibody-like entities as shown in Fig. 1a, but also with a variety of additional format

derivatives, as long as they harbor Fc regions that can be paired with exchange-driving dummies. This includes Fabs, sterically constrained Fab-arms (e.g., Contorsbodies) or other binders such as single-chains, single domains or scaffold binders[1–3,7,8,33,34]. Figure 1b shows the combination of educts with binding entities in various formats, such as entities with Fab arms attached to the Fc N-terminal, C-terminal, or both. Combining such educts, bsAbs are generated that are composed of different binding entities in multiple formats (combination of three input formats with each other, for example, results in nine different output formats for each pre-selected antibody pair). These formats are termed for binderA–binderB combinations according to their mono or duo placement of binding entities at N- or C-termini (Fig. 1b). The N–N format harbors one A- and one B-binding functionality as regular N-terminal Fabs. The C–C format harbors one A- and one B-binding functionality each attached to the C-termini of CH3. N–C and C–N formats harbor one binding specificity as N-terminal Fab and the other C-terminal attached to CH3. N–NC and NC–N formats have binding entities for one specificity attached in N-terminal as well as C-terminal positions, while the binder that covers the second specificity is attached N-terminal. Vice versa, C–NC and NC–C formats have binding entities for one specificity attached in N-terminal as well as C-terminal positions, while the binder that covers the second specificity is attached C-terminal. Finally, NC–NC formats have binding entities for one specificity attached N- as well as C-terminal, and binding entities for the second specificity also in both positions. In summary, the resulting nine formats diversify overall molecular geometry, including paratope distances, number of valencies for each binder as well as molecular size for each antibody pair. Even limited numbers of input molecules, result in large combinatorial matrices. For example, combining just 12 different binders each against 2 different antigens in 3 input formats (requires production of 72 educts) delivers 1296 different binder/format bsAb combinations. Combining 32 binders each (still rather limited number of candidates) in 3 input formats generates >9000 different binder/format bsAb combinations (Fig. 1c). Details of (i) the design of exchange driving dummy interfaces, (ii) educt production and characterization, and (iii) production and characterization of bsAbs generated via dummy-interface driven chain exchange reactions are described in individual sections below.

**Design of dummy-chains and dummy-containing educts.** We desired to have the products of exchange reactions in compositions identical to or at least as close as possible to final lead candidates (screening in final format). Because of that, all alterations that drive exchange reactions and all tags that support purification were placed on the dummy chains. In consequence, bsAb products of exchange reactions harbor regular disulfide-stabilized KiH Fc's. CH3 interface modulating mutations that we introduced into dummy chains were initially designed and modeled on/into existing structures of KiH Fc (PDB 4NQS and 5HY9) without and with the interdomain disulfide bridge (DS), respectively[35,36]. The models were subsequently confirmed by structures of crystallized dummy-containing Fc's (PDB 6YTB, 6YT7, 6YSC, see below). The implemented changes include reverting in CH3 Cys354 and Cys349 (KiH interchain disulfide) back to the parent residues Ser354 and Tyr 349 on dummy knob and dummy hole, respectively. Thus, Fc-dummy heterodimers are flawed in that region because Cys354 and Cys349 on the productive sides are loose ends, not paired with their respective Cys counterparts as they are missing in dummy chains. Additional

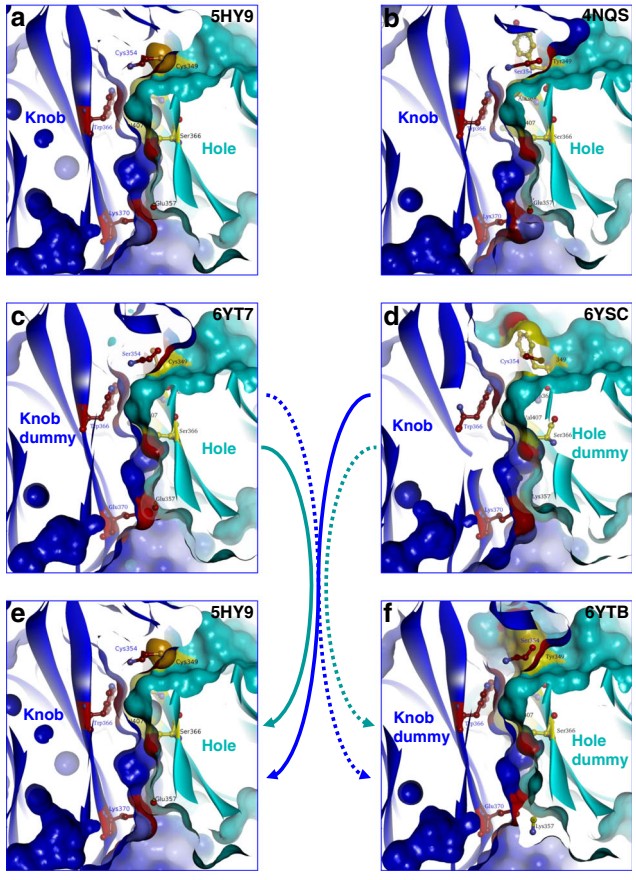

**Fig. 2 Dummy-containing Fc heterodimers with CH3 interface mutations that drive exchange reactions.** CH3 interface mutations that drive the FORCE reaction were originally designed by modifying the existing 5HY9 structure. After successful generation of dummy Fc containing heterodimers, crystals and structures were generated from those Fcs. Shown here are these structures which confirmed the correctness of our initial modeling. **a** 5HY9 structure[36] is characterized by the KiH mutations in ball and stick, an additional DS 354–349, and the salt bridge K370 and E357; **b** crystal structure of the KiH without DS[25,35]; the structures C, D, and F were generated as part of this work and are deposited in PDB ID's 6YTB, 6YT7, and 6YSC. **c** Fc heterodimer with knob-dummy (6YT7); **d** structure of the Fc heterodimer with hole-dummy (6YSC); **e** structure of the interchain disulfide stabilized KiH Fc heterodimer of bsAb products[36]. **f** Structure of the KiH dummy dimer (6YBT).

mutations introduced into the CH3 interface are charge-flip mutations between Glu and Lys at position 357 and 370 of the knob or hole dummies. These residues constitute an interchain salt bridge in regular CH3. The charge flip partially destabilizes the CH3 interface by replacing the 357–370 Glu-Lys salt bridge with equally charged and hence repulsive Lys-Lys or Glu-Glu pairs at those positions. Figure 2 shows the positions of these mutations in crystal structures of the resulting partially flawed Fc interfaces. These structures were generated from crystals of those educt Fc's that harbored the mutated knob or hole dummies, and complementary hole or knob Fc's as described above (Fig. 2c, d). Coordinates for those structures are deposited as PDB 6YT7 (with knob-dummy) and 6YSC (hole dummy), details of the structure determinations are provided in "Methods" and Supplementary Table 1. Figure 2e, f shows molecules with perfect interface complementarity of bsAb Fc (disulfide-stabilized KiH Fc with original salt bridge), and of the dummy-dimer (KiH without interchain disulfide and perfect inverted salt bridge) resulting

from the exchange reaction (PDB 5HY9 for the previously published KiH Cys–Cys[34], and PDB 6YTB for the dummy-dimer deposited as part of this work).

**Production and characterization of educts.** Fc-dummy containing heterodimeric educts were produced via secretion into culture supernatants in the same manner as regular IgGs. All molecules described here were generated via transient transfection of suspension HEK cells (freestyle or Expi), followed by ProtA/SEC purification. Despite harboring partially flawed CH3 interfaces, educts are produced in a robust manner with yields comparable to regular IgGs and without abnormal aggregation tendency. Yields ranged between 40 and 170 mg/L (freestyle) of purified product. The biochemical quality of purified input formats ranges between 90 and 100% monomer on analytical size exlusion chromatography (Fig. 3a and Supplementary Fig. 1). For example, of 48 input molecules generated in 3 different formats, 90% fall into that range and ~10% of input molecules show somewhat reduced homogeneities of 75–90%. All of these input molecules (incl. those with somewhat reduced quality) are suitable for subsequent bsAb assembly reactions because an additional purification step is applied after its completion. Figure 3a shows SEC analyses of proteinA purified material, revealing absence or very low levels of undesired by-products or aggregates. This confirms the benign properties of dummy-containing educt molecules without increased aggregation or undesired homodimerization propensities. Dummy-chain occupation of the CH3 interface overcomes aggregation or dimerization issues sometimes observed with half-antibodies. Correct compositions of purified educts as defined molecules are shown in Fig. 3a. As part of the characterization, and to confirm the predicted effects on the interface structure of the dummy-chain mutations, Fc-dummy knob, or hole heterodimers without attached binder arms were crystallized and their structures resolved (PDB 6YT7, 6YSC, Fig. 2).

**BsAb assembly and purification.** Upon combining two educts, one with knob and one with hole on their productive sides, exchange reactions are initiated by TCEP-mediated limited reduction of hinge disulfides. One hour after initiation, raw reaction mixtures can be subjected without further purification to assays that determine bsAb formation and binding functionality. The results of such bridging-ELISAs performed with a variety of antibody combinations (Fig. 4a) reveal effective formation of bsAbs with desired specificities. Assessment of bsAb formation in raw reaction samples was subsequently applied to define optimized reaction conditions (concentrations of educts and reduction reagent, duration and temperature of incubation), which are shown in Fig. 4b. Initial reaction conditions that we established for all binders and formats were mixture of equimolar amounts at a protein concentration of 0.1–1 mg/ml for each educt in PBS pH 7.4, iniation of the reaction by addition of 25-fold molar excess of TCEP and subsequent incubation at 37 °C for 1–2 h. SEC analyses were applied to monitor and quantify the conversion of educts to bsAbs and dummy dimers (Fig. 4c).

Raw samples of the exchange reaction contain not only the desired bsAb product, but also dummy-dimers as byproducts and potentially some remaining educts or aggregates. The C-terminal His6 affinity tag on dummy chains enables to apply a single robust absorption step to remove all undesired proteins from the completed reaction (concept in Fig. 1a and data in Fig. 3b). Because the productive sides of input molecules lack the tag, properly assembled bsAbs are the only molecules in reaction mixtures that are not retained by affinity matrices (in these examples NiNTA to retain the His6-tag). Examples that

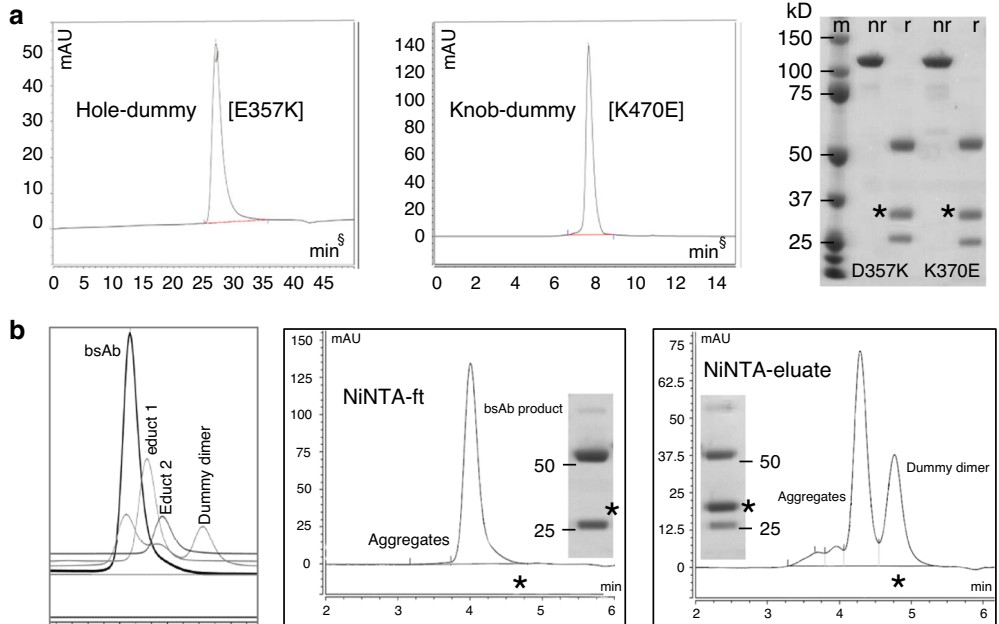

**Fig. 3 Purification and characterization of FORCE educts and products.** Representative examples of multiple experiments (see Section "Methods"— Statistics and reproducibility) show dummy-containing half-antibody-like entities and their conversion to N–N bsAbs. **a** SEC and SDS-PAGE analyses (nr = not reduced, r = reduced) of educts after protein-A purification from culture supernatants. § note that educts and products were initially prepared and analyzed on different SEC runs (explains differences in elution time). After method development, SEC conditions were standardized (comparable elution times in (**b**), see also Fig. 5 and Supplementary Fig. 2). * indicates the engineered dummy chain. **b** FORCE reaction and one-step purification of correctly assembled bsAb. Left panel: SEC differentiates educt and products of the FORCE reaction. Insets show reduced SDS-PAGE (size range between 75 and 15 kDa) of NiNTA flow-through and eluate. *Note the absence of the dummy chain from NiNTA-ft material and its presence in NiNTA-elutions. Source data are provided as a Source Data file.

demonstrate the effectiveness of this purification step and the homogeneity of the purified bsAb preparations are shown in Fig. 3b. We also observed that aggregates became removed by that absorbtion step, because aggregates harbor tag-containing dummies and hence are retained on NiNTA. Although the primary application of this procedure is the combinatorial generation of bsAb binder–format matrices for screening purposes, this approach can also be scaled up, e.g., for rapid resupply of selected molecules. However, we do not currently apply it for drug production.

**Automation of the FORCE process.** Our screening in final format concept requires the generation of large numbers of different bsAbs and formats for subsequent screening to identify those with desired/best functionalities. The FORCE approach delivers large numbers of different bsAbs in different formats with only limited numbers of input molecules required to generate matrices. The large number of resulting bsAbs, however, makes it very difficult to handle FORCE with reasonable effort applying manual wet lab techniques. Process automation overcomes this limitation. Automation requires individual steps of process chains to be simple and robust, prerequisites that are fulfilled for the individual process steps of FORCE. Good expression yields, benign biophysical properties and compatibility with standard (ProtA) purification processes provide the basis for automated generation of the educts. The simple exchange reactions deliver bsAbs with good efficiency, applying one set of conditions for all formats to be generated. Finally, bsAbs can be purified by robust one-step absorbtion-mediated removal of undesired proteins from reaction mixtures. The process-chain and degree of automation that we implied for the individual process steps is summarized in Supplementary Table 2, examples for performance are shown in Fig. 5. A custom-designed liquid handling station covers all

process steps required for transient co-transfection into HEK-Expi cells of expression plasmids encoding the individual chains of dummy-containing educts in 30 ml scale. Supernatants of these cultures are harvested 8 days after transfection and clarified by centrifugation and filtration. Educts are subsequently purified from clarified supernatants via automated small scale proteinA affinity capture on a Tecan Freedom Evo system using robo-columns, followed by preparative size exclusion chromatography on a Dionex HPLC system. Average yields of the automated process for expression/purification of educt molecules are 2–4 mg per 30 ml culture, of sufficient quality (Fig. 5) to serve as input for subsequent exchange reactions. All educts are subsequently set to equal concentration of 1 mg/ml on a Tecan liquid handling system. Exchange reactions are initiated by mixing equal molar amounts (1.5 nanomole of each educt) and by addition of 15 molar equivalents of TCEP in 96-deep well plates and incubated for two hours at 37 °C with agitation. After completion of the reaction, samples are transferred back to the Tecan purification robotic system for affinity-absorption of dummy-containing proteins from the reaction mix which generates the final bsAbs products. SEC analyses confirm successful conversion of precursor molecules to products (Suppplementary Fig. 2), and homogeneity and correct composition of purified bsAbs (Fig. 5).

**Format defines function.** The objective of screening in final format is identification of suitable binder–format combinations, incl. those whose functionalities exceed those of others. FORCE-generated binder–format matrices can identify/differentiate bsAbs with desired functionalities. One example for the relevance of assessing binder–format matrices is shown in Fig. 6. BsAbs were generated that combine different binders recognizing Her-family receptor tyrosine kinases (RTKs) with binders that recognize the death receptor DR5 in different formats. The

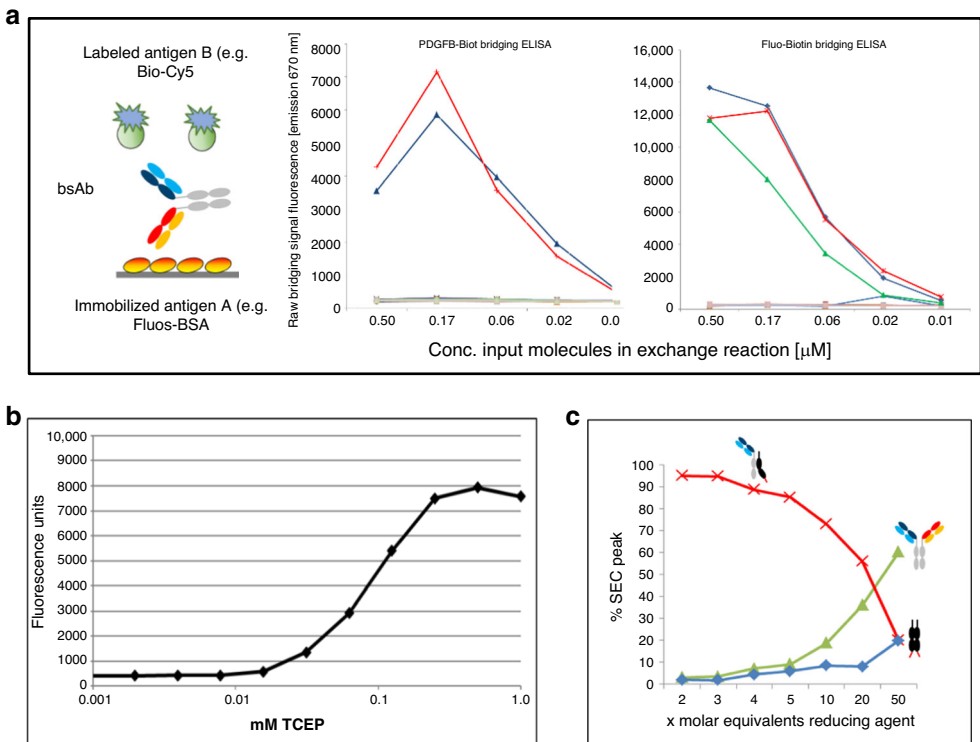

**Fig. 4 BsAb formation can be detected in raw reaction samples via bridging-ELISA and be quantified by SEC. a** Bridging-ELISA and detection of functional bsAbs in raw reaction mixtures without prior purification. Red lines represent hole–knob combinations, blue lines the knob–hole combinations of the respective target antibodies.The green line in the upper right panel is a conventionally produced bsAb control. Additional traces are negative controls (raw reaction mixtures of control antibodies that do not match the antigens used in the respective bridging ELISA). **b** Bridging ELISA indicates that productive exchange requires reduction for initiation. SEC (see also left panel of Fig. 3b) quantification determines TCEP concentrations that enable effective initiation of exchange (TCEP dependent formation of exchange products after 1 h 37 °C monitored by analytical SEC).

desired functionality of these bsAbs is efficient simultaneous binding to RTK's and DR5 on tumor cell surfaces, as prerequisite for triggering apoptosis by DR5-activation on those cells. The automated FORCE process generated a bsAb binder–format matrix to cover combinations of four different RTK binders and four DR5 binders in nine different formats. The four RTK binders subjected to this matrix represented Her1 recognizing V-regions of Cetuximab and Imgatuzumab[37,38] and Her2 recognizing V-regions of Trastuzumab and Pertuzumab[39,40]. The four DR5 binding V-regions that were combined with RTK binders were derived from sequences of Conatumumab, Drozitumab, and Tigatuzumab and KMTR2[41–44]. The capability of bsAbs to simultaneous bind Her1 or Her2 and DR5 was assessed by capturing them on RTK-coated surfaces, subsequently adding labeled DR5 to assess simultaneous DR5-binding. Details of that bridging-ELISA performed in 384-well format are provided in Section "Methods" and Supplementary Figs. 3 and 4. The results of these analyses revealed that for identical target combinations, bsAb binding functionalities depend strongly on chosen binder and format combinations (Methods and Fig. 6). The opportunity to generate a comprehensive binder–format matrix enabled the identification of rules as well as individual particularities for specific Her1 or Her2–DR5 binder combinations.

Commonalities and rules that we deduced included DR5-binders Conatumumab, Drozitumab, and Tigatuzumab to show better RTK–DR5 dual-binding properties with DR5-binding capabilities present at N-terminal positions, compared to exclusive C-terminal DR5-binder placement. Thus, absence of N-terminal DR5-binding functionality (mediated by Conatumumab, Drozitumab, and Tigatuzumab) generates bsAbs with inferior dual binding functionalities (Fig. 6a). In contrast, KMTR2

combined with any Her binder and Tigatuzumab combined with Her2-binders displayed effective dual binding properties irrespective of placement in given formats. Other generalizable observations were reduced co-binding efficacies of bsAbs that harbor only C-terminal Conatumumab or Drozitumab or Tigatuzumab (without additional N-terminal DR5 binder) when juxtapositioned with C-terminal Her binders (Fig. 6b). Increased dual binding efficacy upon bivalent presence of Conatumumab, Tigatuzumab, and KMTR2 combined with all Her binders was also observed (Fig. 6c). One rule that applied without exception to combinations of all Her1−, Her2−, and DR5 binders was efficacy of the N–NC format composed of one N-terminal RTK-binder combined with two N- as well as C-terminal attached DR5 binders (Fig. 6d).

In addition to those general rules, individual particularities and exceptions were observed for combinations of some specific molecules. For example, N–C formats work for combinations of Trastuzumab with all DR5 binders (Fig. 6e, exception to C-terminal DR5-binder interference), and KMTR2 combines well in most format–binder combinations except when combined with Cetuximab in the C–C format. These exceptions underline the importance of FORCE-enabled generation and screening of all possible bsAb binder–format combinations. Figure 6e shows that a definition of binder–format combinations solely by combining rule-deduced best features provides candidates (see methods for details of generating the scoring rubric). However, such analyses/ predictions also generate some false positives (a nuisance which however become eliminated at later steps in bsAb development). A more severe issue of strictly relying on perceived rules are false negatives, i.e., missing efficient functional candidates that are not defined by pre-set rules (Fig. 6f). Comprehensive FORCE-based

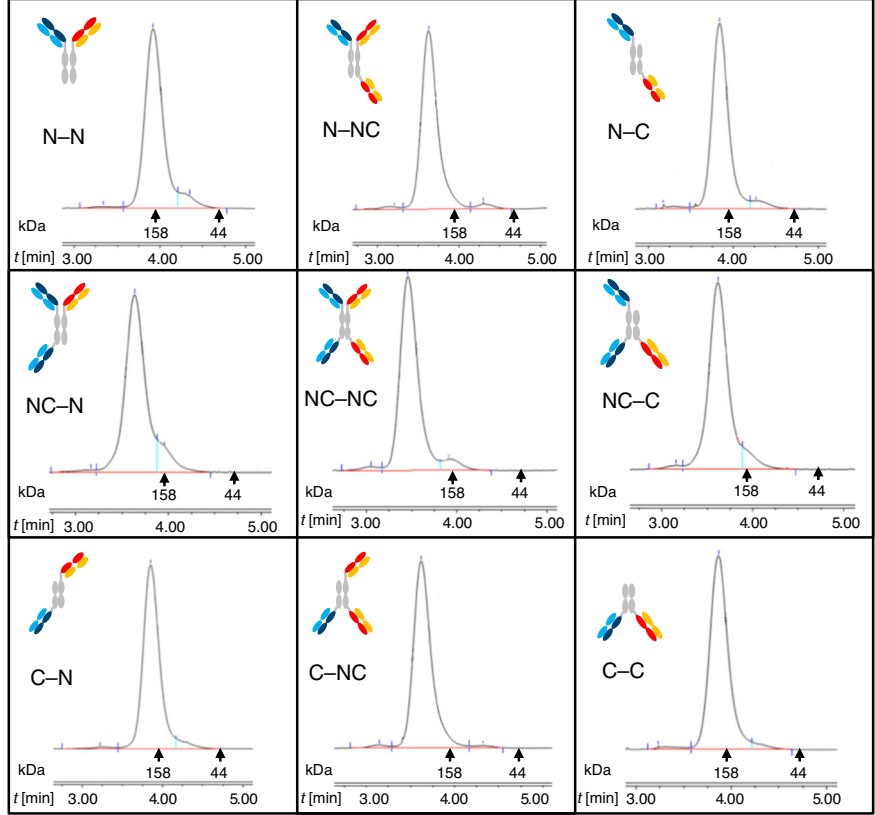

**Fig. 5 SEC analyses confirm homogeneity and purity of bsAb products.** N–N, NC–N, C–N, N–NC, NC–NC, C–NC, N–C, NC–C, and C–C are the different formats generated via FORCE (see Fig. 1b). Elution time of molecular weight standard markers (gamma globulin for 158 kDa, ovalbumin for 44 kDa) is indicated by arrows. Y-axes showing mAU are omitted for clarity (comparable signals and thus yields were obtained for all assembly reactions). Homogeneity of bispecific products (% monomer peak SEC) is: 92% for N–N, 96% for N–NC, 94% for N–C, 87% for NC–N, 92% for NC–NC, 91% for NC–C, 95% for C–N, 97% for C–NC and 96% for C–C. In general, the homogeneities we saw across a large number of bsAb products is suitable for bispecific screening approaches. MS verification is not a standard analytical package for all assembled molecules, but during method development all formats have been verified by MS. Educts (input molecules) were characterized by SEC, capillary-electrophoresis and MS (Supplementary Fig. 1). Source data are provided as a Source Data file.

screens of complete binder–format combination space overcomes these limitations by identifying not only rules that define performing bsAbs, but also those functional entities which do not strictly match those rules.

## Discussion

The FORCE technology bases on a chain exchange reaction of partially repulsive Fc interfaces in heterodimeric antibody derivatives. Those are composed of one productive chain and one dummy chain with and without binding entities, respectively. The exchange reactions are driven by simultaneous presence of mutated CH3 interfaces with complementary (attractive) knob-into-hole mutations and repulsive charge interface mutations in educt molecules. Triggered by reduction of hinge disulfides, two complementary partner molecules spontaneously exchange their chains to generate products with knob–hole as well as additional charge pair attractions in their interfaces. Although harboring partially repulsive interfaces between binder-carrying productive chains and dummies, expression yields and biophysical properties of the FORCE educts are surprisingly similar to regular IgGs. This robustness enables educt production, chain exchange reactions, and product purification by automated processes. Automation of FORCE, and of subsequent assays, becomes essential when increasing the numbers of binders and formats to be combined in matrices because combination spaces grow exponentially with linear increases of the number of input molecules (Fig. 1c).

The output of FORCE approaches is a diverse set of bsAbs that combine different binders in different formats. Despite of this extensive molecule diversity, we observed neither generalizable differences in expression yields or in the quality of different educts formats, nor in pairing efficiencies to generate the products (Supplementary Fig. 1). The only indication for a potentially format-dependent different behavior was with respect to time needed for the reoxididization of hinge disulfides after successful product assembly (assembly reflected by SEC vs. oxidation of interchain disulfides assayed by nonreducing denaturing capillary gel electrophoresis with minor differences observed when applying denaturing analyses too early). We observed that reoxidation takes longer for formats that do not carry Fab-fragments fused to both of their hinge N-termini (N–C, C–N, and C–C formats). However, also for those molecules complete hinge-disulfide reoxidation is achieved by adjusting incubation times after the assembly reaction.

Overcoming the bottleneck of generating large bsAb format–binder matrices enables, when combined with high-throughput functional assays, the identification of best binder–format combinations for desired functionalities. BsAbs with superior features compared to others can in most cases not be obtained by combining two different binders with highest affinities in one preset format. Instead, optimal bsAbs combine binders with compatible epitopes/paratopes and affinities in optimal valencies and format geometries. Large FORCE generated matrices (exemplarily shown

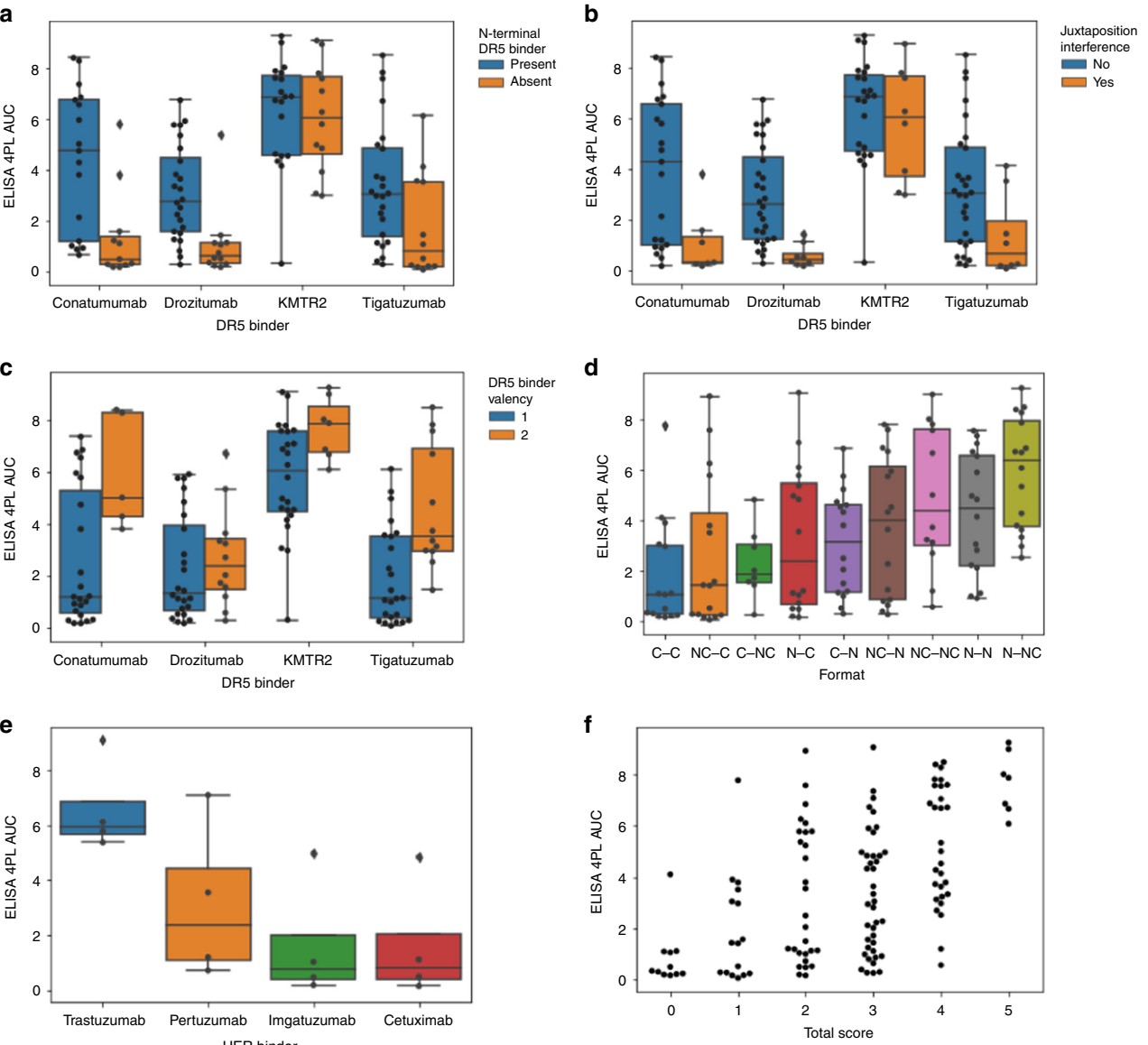

**Fig. 6 Format defines function—the importance of combining suitable binders and formats. a** Reduction of dual binding functionality in bsAbs that lack N-terminal DR5 binding entities. **b** C-terminal juxtapositioned Her binders affect DR5 binding of monovalent C-terminal DR5 binders; **c** Bivalent Conatumumab, Tigatuzumab, and KMTR2 enhances Her-DR5 dual binding; **d** the N–NC format works without exception for all Her–DR5 binder combinations; **e** the N–C format works for Trastuzumab combined with DR5 binders in all formats but is not favorable for other Her binders; **f** screening in final format identifies also combinations that would be "excluded" by strictly rule based approaches. For example, Trastuzumab combined with KMTR2 in C-position show good dual binding despite of bad rule adherence (top entities in column 1–3), and Pertuzumab or Trastuzumab combined with Drozitumab show only marginal dual binding despite of good rule adherence (bottom entities in column 5). Rules applied were (1) DR5 binder present at N-terminus; (2) without C-juxtaposition interference; (3) DR5 valency = 2; (4) HER binder present at N-terminus; (5) DR5 binder = KMTR2. For each construct 1 point was assigned for each fulfilled criterion and the total score compared to the ELISA 4PL AUC outcome (AUC determined from $N = 9$ dose–response sets each, Supplemental Figs. 3 and 4). A description of the scoring rubric applied to rank individual members of the bsAb binder–format matrix is provided in the methods section. Data points represent the values of individual molecules, Boxplots indicate 25th percentile (box bottom border), 75th percentile (box top border), median (horizontal line in box), "minimum" [25th percentile − 1.5 * interquartile range] (bottom whisker), "maximum": [75th percentile + 1.5 * interquartile range] (top whisker), and outliers (dots).

in Fig. 6 for a Her1/Her2–DR5 bsAb matrix) identify such combinations. In addition, FORCE-based screens identify rules that define/differentiate superior and inferior functionalities for desired target combinations (Fig. 6). Such rules further support subsequent optimization efforts once candidates have been identified. In addition to that, FORCE-based screens without any preset format restrictions also identify those rare bsAbs that perform well without adhering to the rules that define the

functionality of most others. These interesting exceptions to the rules underline the importance of FORCE-enabled generation and experimental screening of large bsAb binder–format matrices.

In general, the FORCE approach is not limited to the formats and binding entities shown in this study, provided the entities that are addressed harbor IgG–CH3 domains. Other configurations of productive chains as well as usage of binding modules

different from Fab-fragments are possible, as long as such input formats can be produced in an efficient manner. Such future efforts will expand the combinatorial FORCE binder/format space, and drive the discovery of bi- or multispecific biologics with unique functionalities.

## Methods

**Selection of partially repulsive mutations in dummy-CH3.** Following the invention of the knob-into-hole technology[45,46], several additional and/or alternative approaches to drive directed heterodimerization of H-chains have been devised. Some of these include repulsive (undesired CH3 homodimers)-attractive (desired heterodimers) charge pair modifications in CH3 interfaces. The DD–KK (392,409→356,399) doublet of Gunasekaran et al.[47] and the E357–K370 pair described by Yu et al.[48] are examples among many that apply inverted charge interactions instead of steric knob–hole interactions. To retain good expression of educt molecules while simultaneously introducing exchange propensities, we modeled (BIOVIA Discovery Studio 17R2 (Dassault Systèmes, San Diego) and subsequently combined parts of matching disulfide-connected knob-into-hole interfaces with non-matching (opposing) parts of charge-interfaces. A partial disturbance was introduced into the knob- or hole part by reverting the cysteine that forms an interface disulfide[46] back to its respective wild-type residue. This weakens the CH3 heterodimer interactions to a minor degree without inflicting unfavorable biophysical properties to the molecule. To further de-stabilize to a limited degree and simultaneously drive exchange propensities, we analyzed existing ionic interactions at the CH3 interface of the heterodimeric Fc (pdbcode: 5hy9). There are three salt bridges occurring twice in the pseudo symmetric KiH: from the knob to the hole side, E357–K370, K392/K409–D399, D356–K439 and from the hole to the knob side, K370–E357, D399–K409, K439–D356. As the DS linking the two CH3 domains is formed between the knob C354 and the hole C349, the first three salt bridges are much closer to the DS than the second set. We decided to apply the charge switches apart from the DS. To predict the impact of a charge mutation, we calculated the average percentage of the side chain solvent accessibility. K439–D356 is up to 60% accessible by water molecules; the mutants K439E or D356K will results in double negative or positive charges that will be solubilized apart, making this candidate much less interesting to confer a repulsive effect to the educts. Regarding D399–K409, the average side chain solvent accessibility is low (10%) which makes these positions more attractive. However, the emerging lysine doublet in one educt might result in a too strong repulsive effect due to the close proximity of K392. Inversely, for the second educt, the double negative charge might be compensated by the neighboring K392 so that the repulsive effect might be not strong enough. The candidate charge flip between K370–E357 appeared to be most suited with an average solvent accessibility of the side chains around 15%. We therefore chose the combination of K370E on the knob side and with E357K on the hole side. Educts that combine these mutations (see Fig. 2) still retained good expression yields and benign biophysical properties, yet were found to enable efficient chain exchange upon co-incubation.

**Expression plasmids for antibodies and mutated derivatives.** The plasmids described in this study encode antibody derivatives that are secreted into culture supernatants of transiently transfected HEK293 or HEK-Expi (ThermoFisher) cells. They are composed of a pCDNA3-derived vector backbone and harbor cDNAs encoding H- or L- and/or dummy-chains with a human IgG VH secretion leader peptides under control of the CMV-promoter as previously described[4,24]. The cDNA inserts that encode the different variable and constant regions, interface mutations and format fusions were obtained by gene synthesis, DNA sequencing was applied to assure correctness of all plasmids (Geneart, FRG & Twist Biosciences, USA).

**Expression and purification of antibody derivatives.** All proteins described in this study were produced via secretion into culture supernatants in the same manner as regular IgGs and bsAbs which has been previously described[14]. Expression plasmids that contain CMV-promoter driven expression cassettes for secretion of H- or L- and/or dummy-chains were transiently co-transfected in suspension HEK cells, applying either cells and matching transfection reagents of the human embryonic kidney 293-F cells using the FreeStyle™ 293 system according to the manufacturer's instruction (ThermoFisher/Invitrogen) for individual larger scale purifications. The HEK-Expi system (ThermoFisher) was applied according to the manufacturer's instructions for automated production on our robotics platform. For initial non-automated production, after incubation for 7 days in at 37 °C in humidified 8% CO₂ atmosphere antibody derivatives are present as secreted proteins in the culture supernatants. Supernatants were clarified (3500 g centrifugation and 0.22 μM filtration) and antibody derivatives purified by ProtA (HiTrap™ Protein A HP, GE, 28989336, Boston, MA, USA) affinity chromatography and size exclusion chromatography (HiLoad® 26/600 Superdex® 200, GE, 28989336, Boston, MA, USA). Chain exchange reactions are set up by mixing equimolar amounts of two educts in 1× PBS pH 7.4. Chain exchange is started by addition of 15× molar excess of TCEP relative to educt amounts. Mixtures are incubated for 2 h at 37 °C at 400 rmp. Resulting bsAb products were purified by capturing His-tag containing dummy dimers, aggregates and potentially remaining

educts via NiNTA adsorbtion (His-Trap, GE Healthcare, Sigma-Aldrich). Educts and products of FORCE reactions were analyzed by SDS-PAGE in the presence and absence of 5 mM 1,4-DTE as reducing agent (NuPAGE® Pre-Cast 4–12% Tris-Glycine gels) and subsequently stained with Coomassie brilliant blue. Analytical SEC (Superdex 200 analytical size-exclusion column GE Healthcare, Sweden) in 200 mM KH₂PO₄, 250 mM KCl, pH 7.0 running buffer at 25 °C, and capillary electrophoresis (CE-SDS, Caliper Life Sciences) was applied to assess composition and quality of FORCE-generated products.

**Crystallization, structure determination, and refinement.** Crystallization experiments were performed at protein concentrations ranging from 29.7 to 32.7 mg/ml in vapor diffusion sitting drops at 21 °C. Crystallization droplets were formed by mixing aliquots of 0.1 μl of protein solution with 0.1 μl crystallization screen solution (JCSG + Screen, Procomplex Screen, Qiagen). Crystals grew within 4 days for knob-dummy/hole-dummy out of 20% (w/v) polyethylene glycol 3350, 0.2 M Potassium Formate. The knob-dummy/hole protein crystallized within one day out of 25% (w/v) Polyethylene glycol Monomethyl Ether 2000, 0.1 M HEPES, pH 7.5 and for the hole-dummy/knob protein crystals were obtained 12 h after setup out 15% (w/v) Polyethylene Glycol 4000, 0.1 M HEPES pH 7.0. All crystals were rectangular shaped and belonged to the orthorhombic space group P212121. For data collection, crystals were transferred to mother liquor supplemented with 15% ethylene glycol and cryo-cooled in liquid nitrogen. X-ray diffraction data were collected using a Pilatus 6 M detector at the beamline X10SA of the Swiss Light Source (Villigen, Switzerland). The structures were determined by molecular replacement with the program PHENIX[49] and refined with the programs REFMAC5[50] and phenix.refine[51]. The statistics for X-ray model refinement and data processing are listed in Supplementary Table 1. For representations of PDB structures, Discovery Studio was applied (Dassault Systemes BIOVIA, Discovery Studio Modeling Environment, Release 2017, San Diego, USA).

**BsAb detection and quantification by bridging ELISA.** The bispecific binding functionalities of FORCE-generated bispecific antibodies was assessed by bridging ELISA, capturing bsAbs on an ELISA plate by a first plate-coated antigen followed by sample addition and subsequent detection by a second labeled antigen (Fig. 4a). To assess simultaneous binding of FORCE-generated bsAbs to Her1 or Her2 + DR5, recombinant human EGFR-huFc (R&D Systems, Catalog # 344-ER) or recombinant human Her2-huFc (R&D Systems, Catalog # 1129-ER) were coated on 384 well plates (Nunc® MaxiSorp™) 500 ng/ml in 1× PBS pH 7.4, applying 25 μl/well and incubating at 4 °C overnight. After washing with 3 × 90 μl PBST, wells were blocked (90 μl/well blocking buffer, 1 h room temperature), again washed followed by addition of 25 μl sample at the specified dilution followed by 1 h incubation at room temperature. After washing, 25 μl of biotinylated human DR5 (Acrobiosystems, #TR2-H82E6) was added to each well with an assay concentration of 25 ng/ml. After another washing step, detection was carried out by adding streptavidin POD (Roche Diagnostics, #11089153, 1:4000) followed by a wash and addition of 125 μl/well TMB Substrate. Measurement took place at 370/492 nm.

**Statistics and reproducibility.** Data were assembled and processed by Spotfire (Spotfire Analyst 7.11.0 LTS, ©2007 2017 TIBCO Software Inc., Build date: 5/29/2018) and Microsoft Excel 2016 32 bit. Python (3.6) using NumPy (1.18.4), Pandas (1.0.3) and SciPy (1.2.3.) were used for calculations, plots were visualized using Seaborn (0.9.1). Representative figures (Fig. 3) cover congruent/similar results obtained via multiple independent repetitions (e.g., 9 independent samples in suppl. Fig. 2, 48 samples in Supplementary Fig. 1, >100 screening samples produced and functionally characterized in Supplementary Fig. 3). For automated ELISA-based high throughput screens (Fig. 6 and Supplementary Fig. 3), individual samples were applied per protein and concentration ($N = 1$), with each sample being part of a $N = 9$ dose–response dilution group ($N = 9$, 0.02–100 nM). Determination of individual data points without replicates ($N = 1$) is acceptable for high throughput screening approaches because first, outliers are identified by gross deviations from $N = 9$ dose–response data sets (see Supplementary Fig. 3), second, statistics are thereafter performed based on $N = 9$ dose–response area under the 4PL curve (Supplementary Fig. 4) which reduces the effect of individual measurement outliers, and third, the main basis of subsequent data analysis are groups of samples with shared features which reduces the effect of individual sample preparation errors. The statistics for X-ray data processing and model refinement is described in the Supplementary Table 1.

**Scoring rubric to define well performing Her1/Her2-DR5 bsAbs.** To differentiate functionalities of binder combinations in different formats, blank-corrected ELISA readout data were subjected to four-parameter logistic (4PL) regression with least-squares fit. Individual curves showed high variation in combinations of the bottom asymptote, Hill's slope, EC50 and top asymptote. Therefore, we chose to base comparisons of the different binder combinations in different formats on the area under the 4PL curve (AUC)[52], which incorporates all four parameters (Supplementary Fig. 4). Statistical significance of differences between groups was tested using the Wilcoxon–Mann–Whitney two-sample rank-sum test. Calculations were performed in Python using NumPy, Pandas, and SciPy and plots visualized using Seaborn[53]. Individual parameters that were assessed as potentially influencing

functionality reflected by ELISA outcome were binder combination (which Her-binder combined with which DR5-binder), valency of each binder (monovalent or bivalent), position of each binder (N-term., C-term., N-term + C-term) and geometry (N–N, C–N, NC–N, N–C, C–C, NC–C, N–NC, C–NC, NC–NC format). The results of these analyses reveal incompatibilities, compatibilities, and preferences of individual parameters with bsAb functionality and enabled to identify one best-performing feature for each parameter, namely: (1) DR5 binder present at N-terminus; (2) no juxtaposition interference; (3) DR5 valency is 2; (4) HER binder present at N-terminus; and (5) DR5 binder is KMTR2. To test whether the combination of the best individual features could be used to identify the best-performing constructs (Fig. 6), we defined desired features according to the above described results. For each construct, 1 point was assigned for each fulfilled criterion, resulting in a total score between 0 and 5 for each molecule. The total score of a molecule was then compared to the ELISA 4PL AUC outcome (see Fig. 6).

**Reporting summary**. Further information on research design is available in the Nature Research Reporting Summary linked to this article.

## Data availability

All mutations are described by position and identity and confirmed by X-ray structure determination. The structures that we generated as part of this study are deposited in PDB, ID's 6YTB, 6YT7, and 6YSC. Raw data that form the basis of Her1/2-DR5 bsAb experiments are provided in Supplementary Fig. 3. Source data are provided with this paper.

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

## Acknowledgements

We thank Florian Buddefeld and all members of past and current FORCE-teams for their valuable contributions. C.B. has been supported by the Roche Postdoc Fund (RPF) program.

## Author contributions

S.D. and U.B. devised the concept, initiated, and coordinated the work. K.M., F.B., H.D., B.N., E.H., M.T., and M.W. generated the molecules and/or experimental data and evaluated them. G.G. designed the CH3 interface mutants, U.B., K.M., and S.D. format the variants/combinations; G.G., A.K., and L.L. generated and refined the structures. H.D., B.N., and S.D. devised and applied the automation. All authors processed, evaluated, and interpreted the experimental data. U.B. and S.D. compiled the paper supported by all other authors who contributed to, read, and approved it.

## Competing interests

All authors are employed by Roche, Roche is interested in antibody engineering, therapeutic application of antibody derivatives, and targeted therapies. S.D., K.M., F.B., H.D., E.H., G.G., and U.B. are co-inventors of patent applications related to FORCE technologies. U.B., G.G., F.B., C.B., and S.D. own Roche stock (non-voting certificates).
