## [Peer Review File · Nature Communications]

Reviewers' Comments:

Reviewer #1:

Remarks to the Author:

Dengl, et al. present a platform for the combinatorial development of bispecific antibodies in a broad variety of modalities. Additionally, this method circumvents the need for a common light chain that often plagues the development of a bispecific antibody. The strategy entails generating a series of heterodimers comprised of a Knob-in-hole Heavy chain paired to a relatively unstable 'dummy' Fc. These half IgGs are then combined and subjected to limited disulfide reduction under conditions that destabilize the dummy Fc. This then promotes the formation of full bispecific IgGs in a combinatorial manner that can be performed with reasonable throughput.

The manuscript was very interesting and could be a valuable addition to the literature, but there are a number of issues that should be addressed before publication:

- 1) In general, figures are poorly laid out and it is difficult or impossible to read the axes markings/labels. Some examples of this are in Figure 1, where the axes are illegible for panel A; and while there is what appears to be a molecular weight marker on the protein gel in this panel, there is no annotation for what this marker represents. Another example is Figure 2, where the two graphs in panel A are completely illegible. However, all graphs/figures should be improved as well as the numerous typographical and spelling errors that should also be addressed.
- 2) On page 14, line 298, the authors claim to use 1.5 nanomolar of each adduct in their exchange reaction. This may be a typographical error and they instead mean 1.5 nanomole; please check.
- 3) On page 9 lines 179 and 197, and page 10 line 206, the authors reference PDB ID 5HEY, however this is the structure of a PDZ domain, not a KiH Fc as suggested. This discrepancy must be resolved.
- 4) It is claimed that the SEC data in figure 5 confirm the correct composition of the molecules, however, raw SEC traces merely show the homogeneity (or lack thereof, in some instances) of the molecules. Furthermore, there is no quantification of the SEC traces or comparison to a reference IgG or gel filtration standard to show that the molecules are indeed eluting as monomers. Regarding the confirmation of correct composition, intact mass spectrometry (or other applicable biophysical methods) would be much more informative and should also be pursued.
- 5) Correct PDB accession numbers should be listed for the structures presented in the manuscript/supplemental information. Perhaps these accession numbers are still being assigned?
- 6) In table ST2 it is suggested that 30 ml HEK expressions are performed in 24 well plates. I don't understand how that is possible? Are there multiple wells used for a single transfection as opposed to performing the expression in a single flask?
- 7) On page 11 line 223, the authors reference supplemental figure S2 to make a claim about homogeneity of their molecules, however, this figure does not address that claim. This should be resolved.
- 8) The authors claim that this method is 'screening in final format.' While I see their point, I'd be interested to know if they think this method of production could be implemented on an industrial scale.
- 9) It would be interesting to know if there are generalizable differences in pairing efficiencies or production of various formats (N-N, N-C, NC-N, NC-C, Etc.)
- 10) The authors generate a scoring rubric to identify the features that determine a best-behaving molecule for their given HER2/DR5 example. They should explain this in more detail to further highlight the utility of their method.
- 11) This method seems like it'd be applicable to the generation of tri- or tetra-specific molecules if there is either a common light chain for two of the paratopes or if a single-chain format (e.g. scFv, scFab) format is employed. Has this been investigated?

Reviewer #2:

None

Reviewer #3:

Remarks to the Author:

Review of manuscript NCOMMS-20-18569 June 9, 2020

In this manuscript by Dengl et al, the authors relate to the challenge that, concerning bispecific antibodies, "one format does not fit all". This means that, when starting from a pair of two antibodies that are paired to form a bsAb, the format has a dramatic effect of the performance of the bsAb. This well-accepted fact makes it necessary to evaluate as many different formats as possible early on during bsAb development. The engineering of multiple bsAb format is quite laborious, time and effort consuming.

As a possible solution for making multiple formats of bsAbs, the authors present their elegant solution they named "Format Chain Exchange (FORCE)". According to this approach, 'Input molecules' for the generation of bi/multi-valent bsAbs are monospecific entities similar to knobs-into-holes half antibodies that are paired during initial expression with complementary CH3-interface modulated and affinity-tagged dummy-chains. These dummy-chains contain mutations that lead to limited interface repulsions without compromising the expression or biophysical properties of educts.

Combination of educts triggers spontaneous chain exchange reactions driven by partially flawed CH3-educt interfaces resolving to perfect complementarity. This generate large bsAb matrices with different binding entities in multiple formats. The dummy-chains carry purification tags that, following chain exchange (aided by gentle reduction of the hinge disulfide bonds) provide for the removal of the dummy-chains monomers, dimers and protein aggregates by affinity chromatography, while the correctly-assembled bsAb remains tag-free in the un-bound fraction of the affinity column. The chain exchange is made possible by using in the dummy chains the 1996 version of Knobs-into-holes (KIH) heavy-chain heterodimerization approach, and not the 1998 version that, in addition to the KIH mutations in CH3 contain two engineered cysteines that create an artificial disulfide bond between the CH3 domains. Thus, while the half-antibody-dummy chain adduct does not contain that engineered interchain disulfide, the final bsAbs do (providing additional stability). A "charge flip" – additional mutations in CH3 provide for electrostatic repulsion between the half antibody and the dummy chain, while no repulsion exists in the final bsAbs.

Thus, starting from a limited set of monovalent building blocks, their combinatorial assembly provides a very large of corresponding bsAb formats. For example, starting from 3 formats of monovalent building blocks (to which they refer as "adducts"), 9 bsAbs are created that diversify overall molecular geometry including paratope distances, number of valencies for each binder as well as molecular size for each antibody pair. The fact that the final bsAbs are free of purification tags and the modular nature of the "FORCE" process allows automation as well as "screening in format". The compatibility with automation is critical here, as the large number of resulting bsAbs makes it very difficult to handle FORCE with reasonable effort applying manual lab techniques. Here, I must comment that the authors demonstrated automation by using "A custom-designed liquid handling station covers all process steps required for transient co-transfection into HEK-Exp cells of expression plasmids encoding the individual chains of dummy containing educts in 30ml scale". I can only hope that such a liquid handling station can be assembled by any interested user with little difficulty.

Here, FORCE is demonstrated by screening binder/format bsAbs that simultaneously bind Her1/Her2 and DR5 without encountering binder or format -inflicted interferences.

This is a vary elegant antibody engineering approach that, in my opinion, should be accepted for publication in NCOMMS.

There are minor points that should be corrected as a very minor revision:

- 1) Abstract, line 41: I don't think the word "exemplarized" exists in the English vocabulary. This sentence should be re-phrased.
- 2) Page 6 line 133: . . . "described in detail in subsequent chapters" – replace "chapters" with "sections". Same comment for page 7 line 166.
- 3) I would change the title of Figure 2 from "Figure 2: Dummy-containing Fc heterodimers with CH3 interface mutations drive exchange reactions" to "Figure 2: Dummy-containing Fc heterodimers with CH3 interface mutations that drive exchange reactions".
- 4) page 12 line 251 correct "rea gent" to "reagent".

Dear Reviewers of Nature Communications,

please find enclosed the revised version of our manuscript 'Format Chain Exchange (FORCE) – a robust high throughput approach for combinatorial generation of bispecific antibodies in different formats' by Stefan Dengl et al. We have addressed all topics that were raised and amended/clarified or added to the manuscript as suggested. We thank the reviewers, their comments were valid and constructive and the modifications that we made to address them improved the manuscript. Our point-by-point reply is attached below

Best regards, Ulrich Brinkmann & Co-Authors

Reviewer #1

1) In general, figures are poorly laid out and it is difficult or impossible to read the axes markings/labels. Some examples of this are in Figure 1, where the axes are illegible for panel A; and while there is what appears to be a molecular weight marker on the protein gel in this panel, there is no annotation for what this marker represents. Another example is Figure 2, where the two graphs in panel A are completely illegible. However, all graphs/figures should be improved as well as the numerous typographical and spelling errors that should also be addressed.

→ *We amended the figures for increased clarity, added marker descriptions and increased letter sizes. We spell-checked and proof-read the complete manuscript and corrected typos and language.*

2) On page 14, line 298, the authors claim to use 1.5 nanomolar of each adduct in their exchange reaction. This may be a typographical error and they instead mean 1.5 nanomole; please check.

→ *The reactions contain 1.5 nanomole of each educt. We apologize for the oversight, corrected our mistake accordingly, and re-checked all other numbers/descriptors throughout the manuscript.*

3) On page 9 lines 179 and 197, and page 10 line 206, the authors reference PDB ID 5HEY, however this is the structure of a PDZ domain, not a KiH Fc as suggested. This discrepancy must be resolved.

→ *The PDB ID that we applied for our modeling was 5HY9 (instead of 5HEY). This was corrected in the manuscript text and figure.*

4) It is claimed that the SEC data in figure 5 confirm the correct composition of the molecules, however, raw SEC traces merely show the homogeneity (or lack thereof, in some instances) of the molecules. Furthermore, there is no quantification of the SEC traces or comparison to a reference IgG

or gel filtration standard to show that the molecules are indeed eluting as monomers. Regarding the confirmation of correct composition, intact mass spectrometry (or other applicable biophysical methods) would be much more informative and should also be pursued.

→ *We agree that SEC traces are by themselves not sufficient to prove correct composition. We've included those exemplary traces just to provide examples with the 'most simple' method. We have explained that in the text and added the marker positions to Fig. 5 to show that the molecules are indeed eluting as monomers. We also added information that address additional biophysical methods applied to our molecules to Fig. S1 legend (see also topic 7 below). We also addressed partial inhomogeneities of the traces in the Figure legend. We added figure S2 for further proof of FORCE-mediated conversion of educts to desired products (overlaid SEC elution profiles of smaller educts and increased defined size of FORCE-generated products).*

5) Correct PDB accession numbers should be listed for the structures presented in the manuscript/supplemental information. Perhaps these accession numbers are still being assigned?

→ *The structures that we generated are submitted, PDB-ID's already assigned (6YTB, 6YT7, 6YSC). We have now provided these ID's in the manuscript.*

6) In table ST2 it is suggested that 30 ml HEK expressions are performed in 24 well plates. I don't understand how that is possible? Are there multiple wells used for a single transfection as opposed to performing the expression in a single flask?

→ *Table ST2 provides indeed an overly simplified description. The 30 ml transient HEK expression is performed in 6-deepwell microplates (#CR1406, EnzyScreen, Netherlands) followed by centrifugation and sample transfer into 24-deepwell microplates (#360080, Porvair Sciences, UK) for subsequent purification. Thus, the cleared supernatant product (but not the expression step) of the process step is deposited in 24well plates. By doing so, the 30 ml transient expression supernatant is distributed to 3 wells on the 24-deepwell microplates. We have corrected/added this information incl. volumes and identity of the consumables to the legend of table ST2.*

7) On page 11 line 223, the authors reference supplemental figure S2 to make a claim about homogeneity of their molecules, however, this figure does not address that claim. This should be resolved.

→ *The reviewer is correct, molecule homogeneity is shown in the lower panel of Figure S1 (not in Fig. S2). We have corrected the reference to refer to Fig. S1. We have also expanded the Fig. S1 legend to include a more detailed description of the characterization performed for FORCE-generated molecules (see also response to topic 4 above).*

8) The authors claim that this method is ‘screening in final format.’ While I see their point, I’d be interested to know if they think this method of production could be implemented on an industrial scale.

→ *The major advantage of FORCE is the efficient combinatorial generation of bsAb binder-format matrices, a limited number of input molecules generate output molecules in exponential numbers. This approach can be scaled up, e.g. as we do for rapid re-supply of selected molecules after the screening phase. However, we do not currently consider FORCE as a drug production technology after the desired leads are identified. Instead, we convert the identified binder-format combination to a ‘standard’ (e.g. Crossmab -based) bsAb of identical composition, geometry and format. Thereby, only one stable producer cell line is required (FORCE would need two!) to produce the bsAb already in its final form without an additional assembly reaction. We have now mentioned this aspect in section 4 of our manuscript.*

9) It would be interesting to know if there are generalizable differences in pairing efficiencies or production of various formats (N-N, N-C, NC-N, NC-C, Etc.)

→ *Interestingly we observed neither generalizable differences in expression yields or quality of different educts formats (see Fig. S1), nor in pairing efficiencies to generate the products. The only indication for a potentially format-driven, different ‘behaviour’ was with respect to time needed for the re-oxidization of hinge disulfides after successful product assembly (assembly reflected by SEC vs oxidation of inter-chain disulfides assayed by non-reducing denaturing capillary gel electrophoresis with minor differences observed when applying denaturing analyses too early). We saw that re-oxidation takes longer for formats that don’t have Fab-Fragments fused to both hinge N-termini (especially N-C, C-N and C-C) formats. However, full oxidation is achieved by adjusting incubation times after the assembly reaction. We added these statements and observations to the legends of Figures 5 and S1.*

10) The authors generate a scoring rubric to identify the features that determine a best-behaving molecule for their given HER2/DR5 example. They should explain this in more detail to further highlight the utility of their method.

 *We appreciate this comment because we realize that our description of the scoring rubric appears somewhat ‘hidden’ in the 2nd half of the S2 bridging-ELISA supplement. To explain the scoring rubric in more detail, we now present these analyses as a separate expanded (more explanations & details) section S3 in the data supplement, and refer to this supplement in the main text.*

11) This method seems like it’d be applicable to the generation of tri- or tetra-specific molecules if there is either a common light chain for two of the paratopes or if a single-chain format (e.g. scFv, scFab) format is employed. Has this been investigated?

→ *The technology can indeed be expanded towards multispecificity and additional formats, as long as they harbor Fc-regions that can be paired with exchange-driving dummies. Examples include various types of binders, incl. single-domains, scaffolds, or sterically restrained contorsbodies. The initial set of 3x3=9 molecules/formats that we describe here can after identification as leads be converted for expression in one stable producer cell line (see topic 8 above). This may also be possible for the*

additional formats. Covering all other format possibilities in detail, however, would exceed the scope of this paper. We have briefly mentioned the option to expand towards multispecificity and additional formats, in section 1 of our manuscript.

Reviewer #3

“A custom-designed liquid handling station covers all process steps required for transient co-transfection into HEK-Expi cells of expression plasmids encoding the individual chains of dummy containing educts in 30ml scale”. I can only hope that such a liquid handling station can be assembled by any interested user with little difficulty.

→ *We have added more details of the automated procedure, including source of growth vessels and intermediate containers (see also response to reviewer 1 above). The custom-designed liquid handling station combines different commercially available modules (incl. liquid handling / transport robots, sterile boxing, incubator, harvesting centrifuge, CO₂-incubator, media/buffer/reagent reservoir stations). These were assembled into a transfection robot with custom-designed interfaces according to our specifications by a company specialized in lab-automation. We have added this information to the legend of table ST2.*

Added note: There are several competing companies on the market, all of which are capable to assemble the described system. Examples for ‘usual suspects’ include Beckmann, Hamilton, Tecan and others, just to name a few. We prefer not to name the company that we chose from the usual suspects to assemble our platform as that might serve a potential disadvantage to its competitors.

Minor points:

- 1) Abstract, line 41: I don’t think the word “exemplarized” exists in the English vocabulary. This sentence should be re-phrased. → *we re-phrased the sentence to read ‘... ‘...Examples that demonstrate the relevance of screening binder/format combinations are provided as ...’*
- 2) Page 6 line 133: . . . “described in detail in subsequent chapters” – replace “chapters” with “sections”. Same comment for page 7 line 166. → *the term ‘chapters’ was replaced with sections*
- 3) I would change the title of Figure 2 from “Figure 2: Dummy-containing Fc heterodimers with CH3 interface mutations drive exchange reactions” to “Figure 2: Dummy-containing Fc heterodimers with CH3 interface mutations that drive exchange reactions”. → *we changed the title of Figure 2 as suggested by the reviewer.*
- 4) correct “rea gent” to “reagent”. → *we corrected page 12 line 251 and checked for additional typos throughout the manuscript.*

Reviewers' Comments:

Reviewer #1:

Remarks to the Author:

Dengl et al. have substantially improved the manuscript from their first submission. The study details a combinatorial method to generate bispecific antibodies in a variety of arrangements. From a scientific perspective, this is a particularly intriguing study and the authors have addressed many of the concerns in my initial review. Unfortunately, there remain some issues regarding the figures that lead to significant confusion when attempting to interpret the data as presented.

Here are several examples of where the figures could be improved:

- 1) Figure 3B has clearly been edited to remove numbers from the image, however the remaining number fragments and boxes used to cover these numbers make the image appear to have been improperly manipulated. Furthermore, molecular weight markers are not noted on the PAGE gels in this figure, although they do briefly mention a size range in the legend.
- 2) In the two graphs in Figure 4A, the red and blue traces are identified in the legend, however there are several more traces on these plots that are not identified. What are these data and what does their presentation add to the study?
- 3) In Figure S2 the retention of molecular weight standards is included in the legend, but the retention volumes are nearly impossible to interpret from the figures, rendering this information essentially meaningless.
- 4) The axes labels and units on the graphs in Figure S3 are missing or very difficult if not impossible to interpret.
- 5) There is discussion in the figure legend of Figure 5 that is not discussed in the main text.
- 6) Table ST3 is not referenced in the main text (unless I missed it?) and there is no legend for this table to explain the data therein.